# Hypoplastic Myelodysplastic Syndromes: Just an Overlap Syndrome?

**DOI:** 10.3390/cancers13010132

**Published:** 2021-01-03

**Authors:** Bruno Fattizzo, Fabio Serpenti, Wilma Barcellini, Chiara Caprioli

**Affiliations:** 1Hematology Unit. Fondazione IRCCS Ca’ Granda Ospedale Maggiore Policlinico, 20122 Milan, Italy; fabio.serpenti@unimi.it (F.S.); wilma.barcellini@policlinico.mi.it (W.B.); 2Department of Oncology and Onco-Hematology, University of Milan, 20122 Milan, Italy; chiara.caprioli@ieo.it

**Keywords:** hypoplastic myelodysplastic syndrome, aplastic anemia, immunosuppressive therapy, genomic and molecular landscape, immunological aspects

## Abstract

**Simple Summary:**

Hypoplastic myelodysplastic syndromes (hMDS) represent a diagnostic conundrum. They share morphologic and clinical features of both MDS (dysplasia, genetic lesions and cytopenias) and aplastic anemia (AA; i.e., hypocellularity and autoimmunity) and are not comprised in the last WHO classification. In this review we recapitulate the main clinical, pathogenic and therapeutic aspects of hypo-MDS and discuss why they deserve to be distinguished from normo/hypercellular MDS and AA. We conclude that hMDS may present in two phenotypes: one more proinflammatory and autoimmune, more similar to AA, responding to immunosuppression; and one MDS-like dominated by genetic lesions, suppression of immune surveillance, and tumor escape, more prone to leukemic evolution.

**Abstract:**

Myelodysplasias with hypocellular bone marrow (hMDS) represent about 10–15% of MDS and are defined by reduced bone marrow cellularity (i.e., <25% or an inappropriately reduced cellularity for their age in young patients). Their diagnosis is still an object of debate and has not been clearly established in the recent WHO classification. Clinical and morphological overlaps with both normo/hypercellular MDS and aplastic anemia include cytopenias, the presence of marrow hypocellularity and dysplasia, and cytogenetic and molecular alterations. Activation of the immune system against the hematopoietic precursors, typical of aplastic anemia, is reckoned even in hMDS and may account for the response to immunosuppressive treatment. Finally, the hMDS outcome seems more favorable than that of normo/hypercellular MDS patients. In this review, we analyze the available literature on hMDS, focusing on clinical, immunological, and molecular features. We show that hMDS pathogenesis and clinical presentation are peculiar, albeit in-between aplastic anemia (AA) and normo/hypercellular MDS. Two different hMDS phenotypes may be encountered: one featured by inflammation and immune activation, with increased cytotoxic T cells, increased T and B regulatory cells, and better response to immunosuppression; and the other, resembling MDS, where T and B regulatory/suppressor cells prevail, leading to genetic clonal selection and an increased risk of leukemic evolution. The identification of the prevailing hMDS phenotype might assist treatment choice, inform prognosis, and suggest personalized monitoring.

## 1. Introduction

Myelodysplastic syndromes are a heterogeneous group of clonal disorders affecting the hematopoietic stem cell (HSC), characterized by ineffective hematopoiesis with bone marrow dysplasia and various degrees of peripheral cytopenias [1]. Myelodysplasias (MDS) is typically a disease among aged populations, having an approximate incidence of 3–4/100,000/y, which rises to around 30/100,000/y among patients older than 70 [2]. MDS bears an intrinsic risk of evolution to acute myeloid leukemia (AML), which, on the whole, is estimated to be about 30% but differs according to the International Prognostic Scoring System (IPSS) and the presence of specific somatic mutations [3,4]. The outcome is extremely variable, with median survival ranging from over five years to less than six months, which varies according to the different prognostic risk groups [5] and also to the heterogeneity of distinct clinical and morphological entities. A specific subgroup, representing about 10–15% of patients, is that of MDS with hypocellular bone marrow (hMDS). These cases are defined by a bone marrow cellularity <25% on trephine biopsy or by an inappropriately reduced cellularity for their age in younger patients [6]; however, their diagnosis is still a matter of debate, lacking a clear place in the recent WHO classification. Moreover, the boundaries with normal/hypercellular MDS and bone marrow failures (BMF), including aplastic anemia (AA), still constitute a gray zone. The picture is further complicated by the clinical evidence of cytopenia cases that do not reach the criteria for MDS (too mild cytopenia or too little dysplasia), named isolated cytopenia/dysplasia of unknown significance (ICUS/IDUS), also possibly displaying hypocellularity [1,6]. Finally, recently available techniques allowed the detection of recurrent somatic mutations typical of MDS and AA in patients with ICUS/IDUS and the healthy (not cytopenic) elderly population, namely clonal cytopenia of unknown significance (CCUS) and clonal hematopoiesis of indeterminate potential (CHIP). In this review, we will analyze the available literature on hMDS focusing on clinical, immunologic, and molecular features. We will address the difficulties in the differential diagnosis with other cytopenias/aplastic anemias and myelodysplastic syndromes and discuss whether hMDS deserves to be classified, followed, and treated differently from other MDS.

## 2. Clinical Features

MDS diagnosis is based on the presence of persistent cytopenia (hemoglobin, <100 g/L; absolute neutrophil count, <1.8 × 10^9^/L; platelet count, <100 × 10^9^/L); >10% dysplasia in any hematopoietic lineage, blast excess, or MDS-defining cytogenetic abnormalities (reported in about 50% of patients)). Hypoplastic MDS account for 10–15% of all MDS [7,8,9] and are characterized by bone marrow hypoplasia, a low rate of progression to acute leukemia, and poor response to conventional MDS therapies [10]. As of the FAB and WHO classification systems, most hMDS cases fall into refractory anemia and refractory cytopenias with multilineage dysplasia categories, recently renamed MDS, with single- and multiple-lineage dysplasia [7,8,9,10,11,12]. According to the IPSS risk stratification, they usually belong to the low-risk category [12]. As shown in Table 1, clinical and laboratory markers situate hMDS in the middle of the clinical spectrum between normo/hypercellular MDS and AA. In fact, in comparison to normo/hypercellular MDS, hMDS patients are typically younger (with hMDS being the most common MDS form in pediatric patients), with more severe neutropenia and thrombocytopenia, higher transfusion dependency, a lower percentage of blasts, and fewer common karyotypic abnormalities. Although cellularity easily differentiates hMDS from normo/hypercellular cases, the distinction from AA is harder when only clinical features are considered. Compared to AA patients, hMDS are usually older, show marrow dysplasia, and display more BM blasts and more frequent cytogenetic or molecular alterations. Finally, some conditions typical of bone marrow failures are also present in hMDS. Paroxysmal nocturnal hemoglobinuria (PNH) in particular clones and large granular lymphocytes (LGL) clones (discussed later) are more frequent in hMDS than in normo/hypercellular cases [13,14]. These associations may challenge differential diagnosis since both PNH and LGL chronic expansion may be characterized by peripheral cytopenias: the entity of the clone, together with the presence of hemolytic anemia in PNH and organomegalies in LGL, may help to distinguish the two forms.

## 3. Pathogenesis

MDS pathogenesis is thought to be driven by a selective growth advantage of somatically mutated clonal hematopoietic progenitor cells, and AA is dominated by the immune-mediated destruction of marrow precursors. In hMDS, these features may co-occur, as both genetically driven and immunologic mechanisms affect the stem cell compartment in a vicious circle.

### 3.1. Genomic Landscape

Table 2 recapitulates the main cytogenetic and molecular aspects of normo/hypercellular MDS, hMDS, and AA. Since cytogenetic studies [6,12,15,16] showed no remarkable differences between non-hMDS and hMDS, although the latter more frequently show lower-risk karyotypes, the resolution achieved by molecular approaches may be particularly valuable. High-throughput next-generation sequencing (NGS) characterization represents a powerful tool to detect eventually ontogenetic and prognostic features distinctive of hMDS. However, few studies addressed this specific question, given the rarity of hMDS and the lack of sufficiently large and homogeneous cohorts. An initial study from Huang et al. compared 37 hMDS patients to 152 non-hMDS patients of a single-center Taiwanese cohort, without scoring any significant difference in the prevalence of RAS, AML1, JAK2, PTPN11, and FLT3/ITD mutations [12]. More recently, two single-center studies [15,16] investigated selected gene mutations of known importance in myeloid disorders, comparing their prevalence in hMDS vs. non-hMDS patients. Nahza et al. showed that hMDS was associated with fewer somatic mutations and smaller driver clones compared to non-hMDS (including both normo- and hypercellular MDS). This was confirmed by Yao et al., who reported at least one genetic abnormality (including both cytogenetic and molecular alterations) in 57.0% of the hMDS group vs. 76.2% of the non-hMDS group. Looking at the involved gene mutations, no specific variants were exclusively observed as drivers in hMDS, and the incidence of mutations in spliceosome genes, IDH1/2 [16], and RUNX1, ASXL1, DNMT3A, EZH2, and TP53 mutations [15] was lower in hMDS than in non-hMDS patients.

The largest cohort reported so far included 1261 consecutive adult patients investigated and treated at two major institutions (6). The authors systematically outlined clinical, histopathological, and molecular features of hMDS (n = 278) compared to both non-hMDS (n = 727) and AA patients (n = 136), along with other hypoplastic conditions such as ICUS and congenital bone marrow failures. Genetic profile was explored by testing a panel of 24 genes involved in the pathogenesis of myeloid disorders in 93 hMDS, 239 non-hMDS, and 51 non-neoplastic bone marrow failures. Not surprisingly, the genomic landscape of hMDS resulted in-between AA and non-hMDS in terms of the number of somatic mutations (AA < hMDS < non-hMDS), variant allele frequency (AA < hMDS < non-hMDS), and involved genes. Specifically, when focusing on mutation and comutation patterns previously proven as highly specific for MDS [17], it was confirmed that hMDS show a significantly lower prevalence of mutations in splicing factors and comutation patterns involving TET2, DNMT3A, or ASXL1, compared with non-hMDS, but higher than in nonmalignant bone marrow failures. Nevertheless, the integration of cytohistological and genetic features into a score (called hg-score) enabled the segregation of hMDS patients into two distinct groups: one highly consistent with the profile of myeloid neoplasms and the other more closely resembling nonmalignant bone marrow failure, with no evidence of clonal disease [6]. As such, hMDS more likely represents a mixture of entities along a spectrum rather than a homogeneous in-between category. Altogether, the presented evidence indicates that somatic mutations detected in hMDS are largely shared between AA and non-hMDS, with prevalence and clonal size representing the main difference. Moreover, it is currently difficult to establish the relation of hMDS to germline disorders and CHIP, as the presented studies lack a germline sequence control.

### 3.2. Immunological Features

As summarized in Table 3, a variety of immune cells contribute to the pathogenesis of hMDS, including cytotoxic T cells, regulatory T cells, natural killer cells, B-cells, monocytic cells, mesenchymal stem cells, and mastocytes [18,19,20,21]. Additional findings include the development of paroxysmal nocturnal hemoglobinuria (PNH) clones and the positivity of autoimmunity tests. All these factors may be differently altered in AA and MDS and constitute the immunologic signature of bone marrow failures. In hMDS, immunological features are in-between the typical inflammatory/cytotoxic pattern of AA and the dysregulated permissive pattern of MDS (Figure 1).

#### 3.2.1. Cytotoxic T Cells (CTLs)

In MDS, oligoclonal CTLs are increased [22,23] and inhibit the proliferation of hematopoietic progenitors [22,23,24,25,26,27] by inducing apoptosis via the perforin/granzyme and Fas ligand (FAS-L)/Fas-receptor (Fas-R) pathways [28,29,30,31,32,33]. This inhibition is more marked in hMDS, where Fas-L is increased in T cells [34], and bone marrow precursors overexpress Fas-R [28,34,35]. Although antigen specificity is unknown, specific MDS-related antigens may be present. For instance, patients with trisomy 8 overexpress the Wilms tumor protein 1 (WT1), which is recognized as a neoantigen by CTLs [36,37,38]. Other neoantigens or the overexpression of self-antigens (neutrophil elastase, proteinase 3, and HLA-A2-restricted nonameric peptide) may also be implied [38,39]. Functionally, in low-risk MDS, CTLs are more polarized than in high-risk MDS CTLs, with an increased ratio between CTLs expressing interferon (IFN)-γ and those producing interleukin (IL)-4 [40]. This polarization may result from the persistent stimulation by MDS-related antigens. If CTLs fail to control clone proliferation, their level decreases, and MDS evolves [40]. In AA, CTLs are even more increased than in hMDS and they are mainly polyclonal [23,36]. However, skewed TCR variable regions have been demonstrated, displaying high homology with sequences of viral agents (i.e., Epstein–Barr virus and cytomegalovirus) consistent with a molecular mimicry mechanism [23].

#### 3.2.2. T-CD4-Positive Cells

In MDS, T helper cells (Th) are increased and polarized, similarly to CTLs [7,10,40,41]. Those producing IFN-γ (Th1) are expanded compared to those producing IL-4 (Th2) [20,34]. This polarization is more marked in hypoplastic and low-risk MDS (40). Conversely, other regulatory CD4+ T cells (Tregs) are reduced in number and function, favoring the autoimmune inhibition of myelopoiesis [42,43,44,45,46]. Treg reduction within bone marrow has been related to the degree of dyserythropoiesis [46]. Th1 polarization and Tregs depletion are even more evident in AA, where a reduction of effector memory Tregs is also observed. This proinflammatory/autoimmune phenotype shifts toward a more permissive phenotype in high-risk MDS and AML. In this setting, an increased number of effector memory Tregs, Th17, and Th22 suppressors cells has been demonstrated, which may favor immune escape during MDS progression [47].

#### 3.2.3. Large Granular Lymphocytes (LGL)

LGL expansion may occur as a distinct mono-oligoclonal disease [48] or be associated with other conditions, including myeloid malignancies and autoimmune diseases. Clonality may be difficult to demonstrate, but some genetic lesions have been identified, including STAT3, STAT5b, and cytokine polymorphisms [49]. Interestingly, Jerez and Colleagues recently reported the presence of STAT3 mutations in 7% and 2.5% of MDS and AA, respectively, out of a large cohort of 507 patients [50]. In MDS, STAT3 mutation correlated with lower bone marrow cellularity and a higher frequency of chromosome 7 abnormalities. In MDS and AA, LGL (both CD3+ T-LGL and CD3-CD16+CD56+ natural killer (NK) cells) are increased and may inhibit myelopoiesis through the direct killing and production of IFN-γ and tumor necrosis factor (TNF)α [51,52,53,54]. In hMDS, the proportions of NK-LGL and T-LGL cells are even higher and might be useful in differential diagnosis [14].

#### 3.2.4. B Lymphocytes

In both MDS and AA, lower CD19+ B lymphocyte levels have been found compared to healthy controls [40]. This might be due to a relative decrease due to T-compartment expansion or the impaired maturation of B-cell progenitors. As a matter of fact, CD34+CD19+ progenitors and B-related gene expression have been found decreased in AA and MDS [55,56]. The subset of IL-10 producing B regulatory cells (Bregs) is also decreased in AA [57] compared to healthy individuals, as observed for Tregs.

#### 3.2.5. Autoantibodies

Humoral immunity in MDS is further demonstrated by the presence of autoantibodies in up to 20–34% of patients [58,59,60,61], including antierythrocyte antibodies (up to 10% of cases, particularly in low-risk patients and those with ring sideroblasts), and antiplatelet antibodies (up to 27%) [62]. However, true autoimmune diseases occurred only in about 4–8% of patients [63]. Furthermore, Barcellini et al. demonstrated the presence of antierythroblasts autoantibodies in the bone marrow cultures of 58% of low-risk MDS patients [64]. Interestingly, supernatants from positive cultures induced dysplastic changes in normal BM, stressing the likely contribution of humoral immunity to BMF in MDS.

#### 3.2.6. Monocytes

In MDS, the number and activity of monocytic/macrophage cells have been shown to correlate with the apoptotic rate of bone marrow precursors [65]. MDS monocytes display the upregulation of toll-like receptor (TLR)-4, increased production of inflammatory cytokines, and impaired capacity to engulf the apoptotic dysplastic cells [66]. Consistently, Velegraki et al. showed that lower-risk MDS patients had an increased number of TNFα-producing intermediate monocytes (CD14bright/CD16−, proinflammatory cells) and a lower number of classical monocytes (CD14bright/CD16−, with advanced phagocytic and tissue repair activity) [66]. This occurs also in AA, where TNFα-producing macrophages are increased, stimulating T-cell activation and the production of the pathogenetic cytokine IFN-γ [67,68]. Interestingly, depleting macrophages rescued hematopoietic stem cells and reduced mortality in murine models [68]. Finally, monocytes may favor immune escape and leukemic evolution through the overexpression of CD47, a “do not eat me” signal that inhibits phagocytic activity. Altogether, these data suggest that MDS monocytes have quantitative and functional alterations and contribute to the immune dysregulation associated with the disease.

#### 3.2.7. Mastocytes (MCs)

MCs are increased in both MDS and AA and may act as antigen-presenting cells as well as final immune effectors through the secretion of cytokines (i.e., TNF-α and serine proteases) and direct cell-to-cell interactions [69]. In normo/hypercellular MDS, MCs mainly produce tryptase, a potent mitogen contributing to the typical hypercellular marrow. In hMDS and AA, MCs mainly secrete chymase and granzyme that may induce the apoptosis of bone marrow precursors and digest collagen fibers [70,71], contributing to the hypocellularity and absence of fibrosis observed in these settings.

#### 3.2.8. Mesenchymal Stem Cells (MSCs)

MSCs from MDS and AA patients are numerically and functionally altered compared with those from healthy controls and participate in disease pathogenesis and evolution [72]. In particular, in AA and hMDS, they show a decreased ability to support physiologic hematopoiesis and exhibit a proinflammatory phenotype (type 1 MSCs), thus contributing to bone marrow failure and cytopenias. In high-risk MDS and acute leukemia, MSCs favor tumor escape (type 2) and contribute to the permissive leukemic niche. Similarly, myeloid-derived suppressor cells have been recently shown to be expanded in the blood and marrow of MDS patients, contributing to ineffective granulopoiesis through the production of inflammatory molecules, such as transforming growth factor-beta (TGF-β), nitric oxide, IL-10, and arginase, impairing T-cell surveillance on MDS progression [73,74,75].

#### 3.2.9. Cytokine Dysregulation

The expression and secretion profile of multiple cytokines, such as TNF-α, IFN-γ, TGF-β, IL4, IL6, IL10, and IL17, have been found dysregulated in MDS [7,10,34,76]. TNF-α is highly expressed by CD4+ and CD8+ T lymphocytes from low-risk MDS patients [34,77,78] who display a high frequency of G/A polymorphism at position -308 of the TNF-α promoter [79]. TNF-related apoptosis-inducing ligand (TRAIL) is aberrantly overexpressed in MDS cells, resulting in increased apoptosis [80,81]. IFN-γ is also overexpressed in marrow, blood, and serum, and its blocking in vitro improves marrow colony formation in hMDS patients [34,79]. The inhibition of interferon regulatory factor-1 (IRF-1), frequently inactivated in high-risk MDS, may induce aberrant proliferation of the MDS clone [51,52,82]. Conversely, high IRF-1 levels are associated with favorable prognosis and autoimmune phenomena in MDS [83]. Furthermore, high IL-12 and IL-17 levels have also been described in MDS, further enhancing IFN-γ production [84].

TGF-β and its pathway are potent inhibitors of hematopoiesis, trigger the apoptosis of hematopoietic progenitors, and are upregulated in MDS [85]. Interestingly, the analysis of cytokine polymorphisms in hMDS showed a high TGF-β secretory phenotype and a decreased IL-10 phenotype [79]. Defective IL-10 may allow the enhanced production of TNF-α and IFN-γ responsible for increased apoptosis, typical of hMDS [85,86]. AA is dominated by even higher TNF-α and IFN-γ, but the suppressor cytokine TGF-β and the anti-inflammatory IL-10 are both overexpressed, possibly as a rebound phenomenon [79]. Finally, IL-10 is higher in high-risk MDS, favoring decreased immunosurveillance on leukemic evolution.

#### 3.2.10. PNH Clone

PNH clones are found in 10 to 20% of MDS and up to 60% of AA patients [6,87]. Again, hMDS patients display an intermediate prevalence of PNH clones, about 40% of cases [6,87,88]. Regarding clone size, while AA patients usually present with 20–50% larger clones of granulocytes, MDS usually display smaller PNH populations, generally <10% [87]. Small PNH clones have particularly been shown to correlate with deeper cytopenias and higher LDH levels compared to PNH-negative patients. PNH clones are thought to arise through multistep pathogenesis involving the acquisition of a PIG-A mutation in the HSCs and the subsequent expansion of the mutated clone that is spared by the autoimmune attack. The association of PNH clones with hMDS further highlights the importance of autoimmune activation in bone marrow failures, where the PIG-A mutated clone represents an immune escape.

In summary, genetically driven (MDS-like) pathogenic and immunologic mechanisms (AA-like) both affect the stem cell compartment in hMDS. The prevailing factors may induce the polarization of individual hMDS cases toward either end of the phenotypic spectrum (Figure 1).

## 4. Diagnosis of Hypoplastic MDS

The distinction of hMDS from MDS and AA results from three main features: cellularity on marrow trephine, dysplasia, and cytogenetics. The difference with other MDS is mainly based on cellularity on BM histology, using a threshold between 20 and 30% [11,12,89,90,91]. However, it has been evident that age-related changes should be taken into consideration to avoid missing younger hMDS patients or overdiagnosing elderly patients. Recently, Bono et al. showed that the clinical and prognostic features of younger patients with cellularity higher than 25% but lower than expected for their age, as well as elderly cases with cellularity <25% but normal for their age, were comparable to hMDS with cellularity <25%. In general, cellularity less than 30% under the age of 70 years or less than 20% for patients older than 70 years is the accepted criterion [9]. Distinguishing hMDS from AA is more difficult [8,92,93,94,95,96], implying the presence of dysmegakaryopoiesis and dysgranulopoiesis and the identification of sideroblasts or clusters of blasts [97]. Importantly, mild isolated dyserythropoiesis cannot be used as distinctive features, as is very common also in AA. Chromosomal analysis may help to differentiate these categories and disclose clonality. Cytogenetic abnormalities are found in about 50% of MDS [98] and are mandatory to define specific WHO entities (such as MDS with 5q deletion) as well as IPSS and revised IPSS risk stratifications. However, karyotype analysis is technically impaired in cases of bone marrow hypocellularity, and FISH panels specific for MDS-defining alterations (including chromosomes 3, 5, and 7 alterations) have been developed for this purpose [99]. In a recent large series of hMDS, the presence of del7/7q was shown to aid differential diagnosis with AA [6]. However, other authors described that del7/7q appears to be associated with MDS evolving from AA rather than primary hMDS [100]. Although not yet recommended in clinical routines, characterizing somatic mutations by means of NGS panels might increase diagnostic precision. In a recent study [101], a comprehensive genomic evaluation, including both whole-exome sequencing (WES) and a targeted NGS panel, was performed on 115 patients with bone marrow hypoplasia, resulting in a change of diagnosis in 26% of patients, with a direct impact on treatment choice (disease-specific targeted treatments vs. hematopoietic stem cell transplantation, HSCT), donor selection, and identification of at-risk family members in the case of germline variants. Finally, in the differential diagnosis of hMDS, rarer inherited bone marrow failure (iBMF) syndromes should be considered, particularly in younger patients. It is of great importance to investigate family history, extramedullary manifestations, and past complete blood counts. Moreover, specific molecular alterations (including telomere length and genes associated with BMF with germline predisposition according to the WHO classification (1)) may be identified through targeted sequencing.

In Figure 2, we propose a comprehensive diagnostic algorithm where cellularity, dysplastic changes, and cytogenetics are used to distinguish hMDS from other MDS and AA. Various authors suggested that in cases where the distinction between AA and hMDS is not possible, the patient may be managed as an AA and followed by a close clinical-laboratory follow-up to promptly recognize signs of evolution [102,103].

## 5. Therapeutic Approaches

Hypoplastic MDS are initially treated as low-risk MDS. However, the peculiar pathogenic features may suggest alternative perspectives to be tailored on a case-by-case basis, according to the degree of similarity to MDS and AA.

### 5.1. MDS-Like Treatment

Treatment options for low-risk MDS are mainly directed at controlling cytopenias rather than disease eradication. Anemia is managed with red blood cell transfusions and erythropoiesis-stimulating agents (ESA), with a 60% response rate, particularly in patients with low endogenous erythropoietin (EPO) levels [104]. After ESA failure, transfusions and iron chelation remain the mainstay of treatment except for other agents that are used in specific subsets (i.e., lenalidomide for 5q-syndrome and luspatercept in MDS with ring sideroblasts). In thrombocytopenic hMDS patients, steroids and androgens have been used, with variable response rates, particularly in cases with antiplatelet autoantibodies [9]. Hypomethylating agents, azacitidine and decitabine, are licensed for high-risk patients and have a limited role in low-risk MDS, mainly due to the infectious risk and the potential of worsening cytopenia [105,106,107,108]. However, their use in high-risk cases yields better responses if hypocellularity is present [109]. Moreover, this could be a reasonable option for those hMDS cases with high-risk cytogenetic or molecular features (namely ASXL1, RUNX1, TP53, EZH2, SRSF2, and NPM1 mutations) [110], where immunosuppressive treatment might disrupt immunosurveillance and lead to uncontrolled clonal expansion. In these patients, allogeneic HSCT should also be considered if life-threatening cytopenias are present, age and comorbidities are permissive, and a suitable donor is available. Hypoplastic MDS are usually older and more comorbid, thus HSCT-related morbidity and mortality should be weighed against the potential benefits. Moreover, no prospective studies assessing the role of alloHSCT in this specific patient population are available. In a recent small series [111], 20 consecutive hMDS patients underwent HSCT mainly because of high IPSS-R, showing favorable engraftment rates and survival, and no relapse events with a median follow-up of ~3 years.

### 5.2. AA-Like Treatment

In primary AA, frontline immunosuppressive therapy (IST) with horse antithymocyte globulin (ATG) and cyclosporine A (CyA) is the standard of care, except for patients aged <40 years, for whom HSCT should be pursued. IST induces a durable overall response rate (ORR) in about 60–70% of AA patients in an age-related manner [112]. Much preclinical evidence has shown a reduction of CTLs and Th1 cells and an increase in Tregs close to normal range in AA and MDS after IST. Moreover, the presence of a PNH clone has been established as a favorable predictor of response in both diseases [87]. Table 4 summarizes the most relevant reports of IST use in MDS, reporting the proportion of hMDS where available. On the whole, 207 patients were treated with CyA alone and 264 with CyA plus ATG with an ORR ranging from 24% to 82%, with a trend to better outcomes for the combination therapy. In particular, Passweg et al. [89] conducted a phase III randomized trial of horse ATG plus CyA and reported an ORR of about 30%, favorably associated with hypocellularity. A larger retrospective report [113] of 207 patients from 15 centers across the US and EU treated with IST (mainly ATG plus steroids, 43%) described an ORR of 49%, again associated with hypocellularity (present in 25% of patients). Only one small phase I/II clinical trial [34] was designed specifically for hMDS and evaluated the effect of CyA in vivo and in vitro, showing a parallel reduction of IFN-γ-expressing CD4+ cells along with ameliorated marrow function and increased colony formation. Of note, despite the low number of patients, this trial reported the best ORR (73%). Regarding safety, IST seems manageable in this setting, with infectious complications or serum sickness after ATG reported in 10% and 5%, respectively. Renal toxicity after CyA can be difficult to manage and may cause drug discontinuation [114]. Finally, a single case of AML progression a few months after CyA plus androgen treatment in a young hMDS patient was reported, which was hardly related to CyA itself, and some reports of AML progression exist under androgen treatment [115]. Regarding HSCT, in AA, the risk/benefit ratio is even more age-related than in MDS. Interestingly, one retrospective registry analysis highlighted that low-risk MDS patients receiving HSCT as salvage after immunosuppression showed better outcomes than those for whom it was performed at diagnosis. This might suggest postponing HSCT after IST in eligible hMDS patients [116].

Another interesting agent, used in AA as part of conditioning therapy before HSCT, is the anti-CD52 monoclonal antibody alemtuzumab. The latter gave promising results in MDS in a pilot study, with an ORR of 68%, which rose to 77% in the lower-risk group [131]. However, its hematologic and infectious toxicity may have limited further development in this setting. Contrarily, anti-TNF agents (infliximab and etanercept) and the immunomodulatory anti-vasculogenic drug thalidomide showed disappointing results, with an ORR of around 20%, suggesting the need for better characterization of patients who may potentially respond [119,125,130]. Beyond IST therapy, thrombopoietin receptor agonists (TPO-RA), highly effective in AA either frontline in association with IST or as a single agent in relapsed/refractory patients, have been studied in MDS with variable response rates [136,137,138,139,140,141]. In a very recent phase I/II trial on low-risk MDS [142], eltrombopag was given at the maximal dose of 150 mg/day, inducing an ORR of 44%, significantly associated with marrow hypocellularity. Of note, no progression to AML was observed, and a phase III study on low-risk thrombocytopenic MDS cases is ongoing [138]. Given the good results observed in AA [143], some authors also tried androgens in MDS with promising outcomes [144,145] and reported an ORR around 60–70%. In a small case series, four out of six hMDS patients showed at least a partial response, and the outcome was even better than for patients treated with IST [9].

Finally, the new agents targeting the immune microenvironment of MDS and AML need to be mentioned. The latter include the checkpoint inhibitor sabatolimab, which targets TIM3, an inhibitory receptor that regulates adaptive and innate immunity. Sabatolomab is able to restore the antileukemic immune response in MDS and AML [146]. Another molecule is the monoclonal antibody Hu5F9-G4 (5F9) against CD47, a macrophage immune checkpoint and “do not eat me” signal, whose blockade induces tumor phagocytosis and eliminates leukemia stem cells [147].

## 6. Clonal Evolution

Patients with hMDS eventually progress to AML at a five-year rate of around 10–40%, somewhat in-between AA and normo/hypercellular MDS [6,11,148]. Such an event has been correlated to genetic features so that hMDS patients carrying poorer-risk cytogenetic (del7/7q, complex karyotype) and molecular lesions (spliceosome, cohesin, DNA methylation, TP53) show a higher incidence of leukemic transformation [149,150]. The risk also appears age-related, consistently with an age-dependent distribution of somatic mutations, as described in healthy subjects with age-related clonal hematopoiesis (ARCH) [151] and as a result of genomic instability. This is in line with the recent demonstration of altered DNA damage repair pathways in the pathogenesis of hMDS [152,153,154,155]. The proinflammatory immune system of hMDS, far from being an innocent spectator, may fuel clonal progression: proinflammatory cytokines and proapoptotic mediators may accelerate the apoptotic rate of BM progenitors and favor the selection of somatically mutated clones with proliferative advantage [156,157].

## 7. Survival and Prognostic Factors

The first reports of a better clinical outcome in hMDS patients compared to non-hMDS emerged more than 30 years ago [91,158]. Although prognosis in these cases can be correctly predicted by both IPSS (12) and IPSS-R risk stratification models [15], hypocellularity per se has been confirmed as a good prognostic factor in several studies, with median survival ranging between 33 and 58 months in hMDS compared to 19–28 months in non-hMDS [9,159]. Importantly, in two large cohorts [148,159], the prognostic impact of hypocellularity was shown to be independent of both IPSS [12] and IPSS-R [15]. Moreover, BM cellularity further refined the identification of patients with better prognosis among those with low-risk disease. On the contrary, in a more recent nationwide Japanese report [148], BM cellularity did not affect the clinical outcome in high-risk IPSS patients. In the same study, the 66% five-year OS in hMDS patients paralleled the 63% five-year progression-free survival, significantly higher than that of the normo/hypercellular counterpart (five-year OS of 49% and five-year PFS of 41%, respectively) and translated into a high proportion of deaths due to bone marrow failure itself. Of note, 5yPFS to AML rose to 92% when considering only lower-risk patients, in line with data from Bono et al. [6]. Not surprisingly, according to the few existing studies directly comparing hMDS to AA in regard to survival and progression to AML, hMDS seems to stay in-between AA and normo/hypercellular MDS, consistently with its overlapping clinical and pathogenetic features [11]. Given such ambiguity, some efforts have been made to identify predictors of survival and AML progression. For instance, a dedicated risk score was designed in 2012, including clinical parameters (poor performance status, Hb < 10 g/dL, serum lactate dehydrogenase >600 IU/L) as well as cytological criteria (bone marrow blasts ≥5%) and cytogenetics (presence of −7/7q or complex karyotype) [160]. In two studies [15,16], the detection of mutations of known prognostic importance in myeloid neoplasms (i.e., spliceosome genes, RUNX1, ASXL1, DNMT3A, EZH2) did not impact the survival of hMDS, with the meaningful exception of TP53 mutations. Finally, the integrated hg-score from Bono et al. [6] enabled the identification of two groups of hMDS with a significantly different OS and risk of blast progression, and the impact of specific mutations was not reported.

## 8. Conclusions

Although the rarity of hMDS and its uncertain diagnostic boundaries limit data interpretation and definitive conclusions, peculiar pathogenesis is suggested by much clinical, pathological, and molecular evidence. Through various overlap, two different phenotypes of hMDS emerge (Figure 2): one featured by inflammation and immune activation, more similar to AA, where increased cytotoxic cells and their cytokines and decreased regulatory cells favor bone marrow failure; and the other, resembling MDS, where regulatory/suppressors cells prevail and prompt a switch toward a tumor permissive/facilitating milieu. These latter cases are dominated by genetic clonal selection and an increased risk of leukemic evolution. In clinical practice, the identification of the prevailing hMDS phenotype might assist treatment choice (AA- vs. MDS-like therapies), inform prognosis, and suggest personalized clinical monitoring/schedule of re-evaluation.

## Figures and Tables

**Figure 1 cancers-13-00132-f001:**
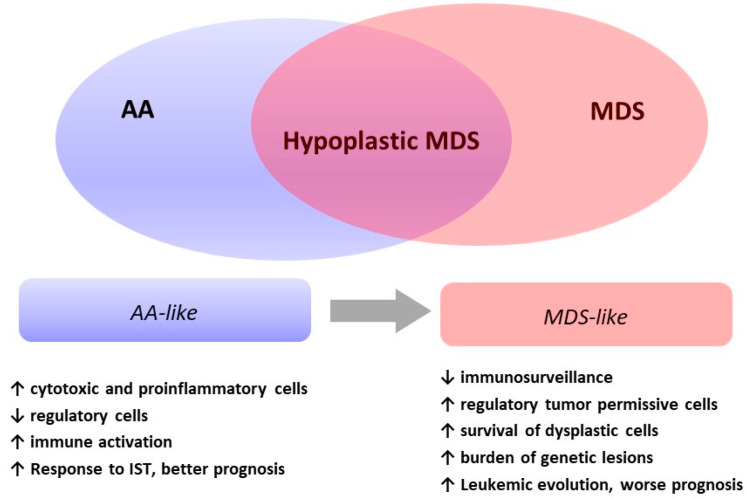
The two phenotypes of hypoplastic myelodysplastic syndromes (hMDS). Clinical features and immunologic and molecular studies identify two phenotypes of hMDS: one with prevailing inflammation and immune activation (increased cytotoxic T cells, CTLs, and T helper 1 cells, Th1, mastocytes, and proinflammatory monocytes; decreased T and B regulatory cells, Tregs and Bregs; type 1 mesenchymal stem cells, MSCs; and higher prevalence of autoantibodies and PNH clones) that we defined AA-like and the other dominated by genetic lesions, clonal selection, and leukemic evolution, named MDS-like. The evolution from one phenotype to the other is marked by a progressive decrease of proinflammatory/proapoptotic immune effectors and an increase of regulatory suppressive cells as well as a shift from type 1 (proinflammatory) to type 2 MSCs (tumor-facilitating), which enable clone selection and tumor escape.

**Figure 2 cancers-13-00132-f002:**
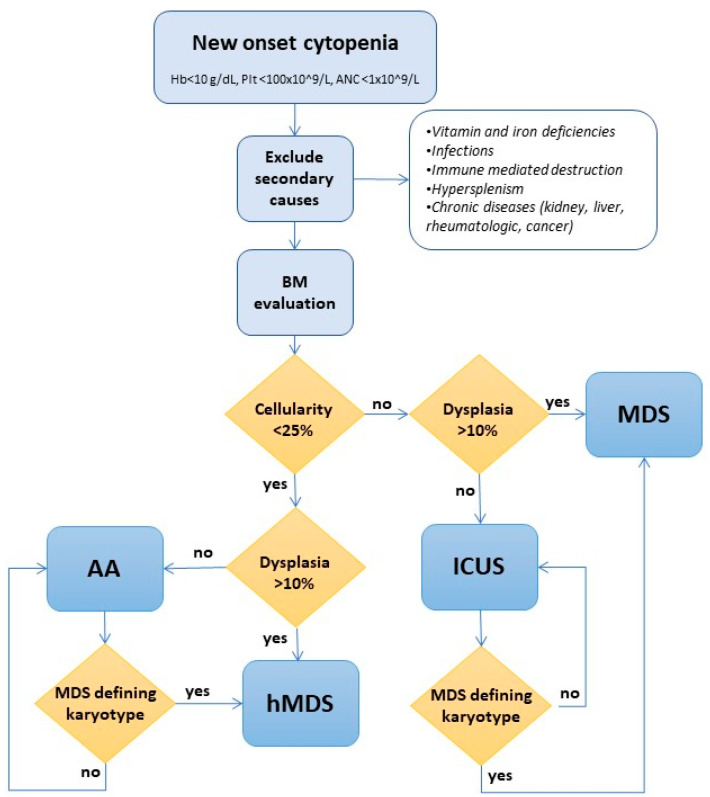
Diagnostic algorithm for hypoplastic MDS (hMDS). The first step is to distinguish “peripheral” from “central” cytopenias based on clinical features such as signs and symptoms of hemolysis or increased catechesis (i.e., splenomegaly) and laboratory features as reticulocytosis, direct antiglobulin testing, and antiplatelet autoantibody positivity (signs of peripheral immune destruction typical of autoimmune hemolytic anemia and immune thrombocytopenia). Subsequently, secondary forms (infections, toxic, cancer, etc.) have to be excluded, and the bone marrow biopsy would separate hypocellular cases from normo/hypercellular cases. The presence of >10% dysplasia of one lineage would distinguish hMDS from AA in hypocellular cases and MDS from idiopathic cytopenia of unknown significance (ICUS) in normo/hypercellular cases. In both hypo-, normo-, and hypercellular cases, if MDS-defining cytogenetics are present, the diagnosis would be hMDS or MDS (according to cellularity) even without dysplasia. Moreover, in hypocellular cases, family history and physical examination will orient the diagnosis of inherited bone marrow failures (iBMF).

**Table 1 cancers-13-00132-t001:** Semiquantitative comparison of clinical and laboratory features among hypoplastic myelodysplastic syndromes (hMDS), aplastic anemia (AA), and low-risk (LR) MDS. LDH, lactate dehydrogenase; PNH, paroxysmal nocturnal hemoglobinuria; LGL, large granular lymphocytes.

Properties	Normo/Hypercellular MDS	hMDS	AA
**Clinical Features**			
Median Age	+++	++	+/−
Male/Female Ratio	>1	=	=/<1
Bleeding	+/−	+	++
Transfusion-Dependence	+/−	+	++
Infections	+/−	+/−	++
**Laboratory Features**			
Cytopenia and Macrocytosis	+	++	++
LDH	+/−	+	++
BM Blasts	=/+	−	−−
Cytogenetic/Molecular Alterations	++	+/−	rare
**Associated Conditions**			
PNH Clone	+/−	+	++
LGL Clone	+	++	+/−
Extrahematologic Autoimmunity	−	++	+/-
**Prognosis**			
Leukemic Evolution	+	+/−	−
Survival	−−	+/−	+/−

**Table 2 cancers-13-00132-t002:** Semiquantitative comparison of genetic features among hypoplastic myelodysplastic syndromes (hMDS), aplastic anemia (AA), and normo/hypercellular MDS. VAF, variant allele frequency.

Mutations	Normo/Hypercellular MDS	hMDS	AA
**Cytogenetic Abnormalities, *n***	++	+/−	rare
**Cytogenetic Abnormalities, Type**	Low to high risk	Mostly low risk	Mostly low risk
**Somatic Mutations, *n***	++	+/−	+/−
**Somatic Mutations, Clone Size (VAF)**	+++	++	+
**Somatic Mutations, Type**			
**Splicing SF3B1, SRSF2, U2AF1, ZRSR2**	+++	+	+/−
**DNA Methylation DNMT3A, TET2, IDH1, IDH2**	++	+	+/−
**Chromatin Modification ASXL1, EZH2, KDM6A**	++	+	+/−
**Cohesin STAG2**	+	+/−	rare
**Tumor Suppressor TP53**	+	+/−	rare
**Signaling CBL, FLT3, JAK2, KIT, NRAS, KRAS**	+/−	+/−	rare
**Transcription RUNX1, CEBPA, ETV6, GATA2, NPM1**	RUNX1 =/++; others = +/−	+/−	rare
**Pathogenic germline RTEL1 mutations**	−	+/−	+

**Table 3 cancers-13-00132-t003:** Immunologic mechanisms involved in the pathogenesis of hypoplastic myelodysplastic syndrome (hMDS), normo/hypercellular MDS, and aplastic anemia (AA). Th, T helper; Tregs, T regulatory cells; LGL, large granular lymphocytes; Bregs, B regulatory cells; IST, immunosuppressive therapy; Th17, T helper 17; NK, natural killer cells; TLR, toll-like receptor; PNH, paroxysmal nocturnal hemoglobinuria.

Types	Normocellular/Hypercellular MDS	hMDS	AA	References
**T-cytotoxic cells (CTLs)**	Increased and oligoclonal. In high-risk patients, IFN-γ-producing CTLs decrease favoring leukemia evolution.	Increased and clonal; produce interferon-gamma (IFN-γ) and decrease after response to IST.	Highly increased and polyclonal. Higher in patients not responding to IST and at relapse.	[14,22,23,24]
**T-CD4+ cells Th and Tregs**	Increased T regs collaborate in the suppression of immune surveillance and leukemic evolution.	Increased and polyclonal Th cells producing IFN-γ. Tregs are reduced and correlate with dyserythropoiesis.	Reduced Tregs and effector memory Tregs.	[23,25,26]
**LGL clones**	Increased polyclonal and oligoclonal (both T-LGL and NK-LGL) more than in AA.		Increased in AA; STAT3 mutation correlates with better response to IST.	[14,27,28]
**B-cells and autoantibodies**	Reduced B-cells and B-related gene expression, possibly recovering after therapy. Autoantibodies are present in up to 20–34.4% of patients) but true autoimmunity only in 4%.		B-cells and IL-10 producing B-regs are reduced and correlate with severity and response to IST.	[14,28,29,30,31,32,33,34]
**Macrophages**	Increased number and activity and upregulation of TLR4, correlating with apoptosis. Increased number of TNFα producing intermediate monocytes (CD14bright/CD16−, proinflammatory cells). Impaired capacity to engulf the apoptotic dysplastic cells.		Increased TNFα producing macrophages. Their depletion induced recovery and reduced mortality in murine models.	[35,36]
**Mastocytes**	Increased tryptase-producing-MCs; tryptase is a mitogen that contributes to hypercellularity.	Increased chymase-producing MCs. Chymase induces apoptosis.	Increased polyclonal benign MCs. Their persistence after IST correlates with poor outcomes.	[37,38]
**Mesenchymal stem cells (MSCs)**	Reduced expression of immunomodulatory cytokines. MSCs show an ineffective production of osteopontin, angiopoietin, Jagged1, and stromal-derived factor 1-CXCL-12, failing to support hematopoiesis. MDS-MSCs display genetic abnormalities associated with the 5q- syndrome or with a high-risk karyotype.		Reduced angiogenic and osteogenic potential, and adipogenicity is increased. AA-MSCs contribute to Treg/Th17 imbalance.	[39]
**PNH clone**	Present in 20% of patients and correlates with better survival and response to HSCT	Present in up to 40% of patients and correlates with higher LDH levels, deeper cytopenias, better response to IST, and survival.	Present in up to 60% of patients and correlates with higher LDH levels, better response to IST, and better survival.	[6,40]
**Cytokine levels**	Proinflammatory cytokines (IFN-γ and TNFα) are increased. In high-risk patients, IL-10 is increased and contributes to the suppression of leukemic evolution.	Proinflammatory cytokines and TGF-b are increased inducing bone marrow failure. IL-10 is reduced and fails to suppress inflammation.	Proinflammatory cytokines and TGF-b are highly increased. IL-10 is increased as a rebound effect.	[7,10,22,41,42,43,44,45,46,47,48,49,50,51,52,53]

**Table 4 cancers-13-00132-t004:** Available studies on the use of immunosuppressive therapy (ATG, antithymocyte globulin, CyA, cyclosporin A, and others) in hypoplastic myelodysplastic syndrome (hMDS). N, total absolute number of treated patients; hMDS%, percentage of hMDS patients among the total number; ORR overall response rate for the whole cohort of patients. “-” = not reported.

Reference	*N*	Study Design	Treatment	hMDS %	ORR	Time to Response (m)	Response Duration (m)
[117]	25	Phase II trial	ATG	-	44	-	10
[118]	17	Retrospective	CyA	53	82	3	-
[119]	83	Pilot study	Thalidomide	15	19	4	10
[120]	61	Phase II trial	ATG	38	34	2.5	36
[34]	11	Phase I/II trial	CyA	100	73	2.3	58
[121]	32	Phase II trial	ATG + CyA	-	26	2.5	12
[122]	30	Pilot study	ATG	27	33	-	15
[123]	50	Retrospective	CyA	20	60	1.8	-
[124]	15	Phase II trial	ATG + etanercept	7	46	-	24–36
[125]	37	Pilot study	Infliximab	-	22	-	6-12
[126]	35	Phase II trial	ATG	11	34	3	9
[114]	19	Phase II trial	CyA	21	58	2.5	-
[127]	25	Phase II trial	ATG + CyA	20	24	2	7
[128]	129	Retrospective	ATG/CyA/ATG + CyA	33	30 (24/8/48)	4	36
[129]	15	Phase II trial	ATG + CyA	-	33	3.7	-
[130]	25	Phase II trial	ATG + etanercept	-	56	2	5–36
[131]	31	Phase I/II trial	Alemtuzumab	35	68	3	-
[89]	45	Phase III trial	ATG + CyA	20	29	-	16
[132]	37	Phase II trial	CyA + thalidomide	14	57	1.8	22
[133]	71	Phase II trial	CyA	48	77	1.5	24
[134]	24	Phase II trial	ATG + CyA	-	25	4	-
[135]	66	Retrospective	ATG/CyA/ATG + CyA	-	42	-	12
[113]	207	Retrospective	Any	26	49	2.5	20

## Data Availability

Not applicable.

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
