# Peer review of "Hypoplastic Myelodysplastic Syndromes: Just an Overlap Syndrome?"

_cancers, 2021, doi:10.3390/cancers13010132_

Round 1

Reviewer 1 Report

The authors describe different features of Hypoplastic myelodisplastic syndromes.

The article is well written and give important suggestions on these hematologic syndrome.

The sections: Introduction, clinical features, pathogenesis with the related tables are very clear and not need any corrections

The section: immunological features and table 3 is not clear and confusing. please correct both.

In the section therapeutic approaches please cite the new insights (if available) on the treatment fo MDS syndromes.

Author Response

We thank the Referee for the careful revision and for the positive comments. We agree that the section on immunological features is complex in nature. As suggested, we summarized main concepts in shorter sentences and rewrote table 3.

As suggested, a sentence was added on the development of new agents targeting the immunologic microenvironment in MDS.

“Finally, the new agents targeting the immune microenvironment of MDS and AML need to be mentioned. The latter include the checkpoint inhibitor sabatolimab, that targets TIM3, an inhibitory receptor that regulates adaptive and innate immune responses, and restores anti-leukemic immune response (Brunner et al, ASH 2020, https://doi.org/10.1182/blood-2020-136855); and the monoclonal antibody Hu5F9-G4 (5F9) against CD47, a macrophage immune checkpoint and “don’t eat me” signal, whose blockade induces tumor phagocytosis and eliminates leukemia stem cells (Sallman et al, ASCO 2019, doi/10.1200/JCO.2019.37.15_suppl.7009).”

Reviewer 2 Report

Dear Authors, I’ve read the manuscript with interest. Hypoplastic-MDS (h-MDS) represents de facto an obscure clinical and pathological entity (In fact h-MDS is not described among the revised 2016 WHO classification of myeloid neoplasms) with many overlapping characteristics shared with other hematologic diseases (BMF, including aplastic anemia; and MDS). Helping physicians differentiating the diagnostic features of it could be relevant from a clinical point of view. As already described previously (Malcovati L, Galli A, Travaglino E, et al. Clinical significance of somatic mutation in unexplained blood cytopenia. Blood. 2017;129:3371–8; Bono E, McLornan D, Travaglino E, et al. Leukemia. 2019 Oct;33(10):2495-2505), there are two distinct phenotypes of the disease, each one with distinctive pathological and clinical features, harboring different prognostic impacts. I have only few comments, below: . I do think Table 1, although interesting from a clinical perspective, may be ameliorated in terms of aesthetics (perhaps a bit too long and not so consistent). . Fig.1 – probably too much info on it; You’ve already described it into the legend (or, may you want to include only specific predominant characteristics for each one of the two clinical entities?) . Table 3 - You can probably omit it as you have already described in detail the immunologic features specifically attributed to single pathologic entities. . Fig.2 – among secondary causes, what do you mean with “immune-mediated destruction”?. May you can switch from AA to BMF, as you already mentioned it among the legend? . Table 4 – I suggest you to include clinical trials only. You also could mention the reference number instead of reporting Authors name and year of publication of each analysis. Best,

Author Response

I do think Table 1, although interesting from a clinical perspective, may be ameliorated in terms of aesthetics (perhaps a bit too long and not so consistent).

We thank the Referee for the thorough revision. As suggested, we shortened table 1 and ameliorated its aesthetics.

Fig.1 – probably too much info on it; You’ve already described it into the legend (or, may you want to include only specific predominant characteristics for each one of the two clinical entities?).

We agree with the Referee and removed some redundant information from the figure itself if already explained within the legend.

 Table 3 - You can probably omit it as you have already described in detail the immunologic features specifically attributed to single pathologic entities.

We agree with the Referee that some information is redundant. We tried to further summarize the table content and, if possible, we would keep it to summarize main concepts.

Fig.2 – among secondary causes, what do you mean with “immune-mediated destruction”?. May you can switch from AA to BMF, as you already mentioned it among the legend?.

We thank the Referee for the comment. By immune mediated destruction we meant autoimmune cytopenias such as AIHA, ITP and AIN. This has been now specified within the figure 2 legend. If possible, we would like to keep AA instead of BMF as to better distinguish it from hypoplastic MDS which is also considered a BMF by many Authors.

Table 4 – I suggest you to include clinical trials only. You also could mention the reference number instead of reporting Authors name and year of publication of each analysis.

We thank the Referee for the comment, and as suggested we removed Authors’ names and added the reference number to the table (now table 3). If possible, we would keep retrospective studies since in these rare forms they can add the value of real life observations and longer follow up.

Reviewer 3 Report

First of all, please proofread the whole manuscript. At least correct the grammar errors in the writing. Please try not to re-use the same sentences in the abstract and the introduction. Please try to make the details (font, style, color, letter case) consistent in all your tables or figures.

The main thoughts are well structured, but the manuscript needs to be further polished.

Author Response

First of all, please proofread the whole manuscript. At least correct the grammar errors in the writing. Please try not to re-use the same sentences in the abstract and the introduction. Please try to make the details (font, style, color, letter case) consistent in all your tables or figures.

The main thoughts are well structured, but the manuscript needs to be further polished.

We thank the Referee for the positive comments. As suggested, we proofread the manuscript and corrected misspellings. As regards tables/figures details, we reshaped figures fonts for consistency. 

Round 2

Reviewer 3 Report

Ok, thanks for the quick response to improve the manuscript.